# Hybrid-EDL: Improving Evidential Deep Learning for Uncertainty Quantification on Imbalanced Data

**Tong Xia**[1]  **Jing Han**[1]  **Lorena Qendro**[1,2]  **Ting Dang**[1]  **Cecilia Mascolo**[1]

**[1] University of Cambridge, [2] Nokia Bell Labs, Cambridge (UK)**
tx229@cam.ac.uk

## Abstract

Uncertainty quantification is crucial for many safety-critical applications. Evidential Deep Learning (EDL) has been demonstrated to provide effective and efficient uncertainty estimates on well-curated data. Yet, the effect of class imbalance on performance remains not well understood. Since real-world data is often represented by a skewed class distribution, in this paper, we holistically study EDL, and further propose Hybrid-EDL by integrating data over-sampling and post-hoc calibration to boost the robustness of EDL. Extensive experiments on synthetic and real-world healthcare datasets with label distribution skew demonstrate the superiority of our Hybrid-EDL, in terms of in-domain categorical prediction and confidence estimation, as well as out-of-distribution detection. Our research closes the gap between the theory of uncertainty quantification and the practice of trustworthy applications. Code is available at https://github.com/XTxiatong/Hybrid-EDL.git.

## 1 Introduction

Uncertainty estimation is key to safety-critical applications like healthcare and autonomous driving, as it allows a deep learning model to know "what is unknown" [15, 30]. Making decisions under uncertainty makes deep learning more trustworthy in the real world [3, 30].

Commonly used uncertainty estimation approaches include deep ensembles and Bayesian neural networks [9]. However, they are either computationally expensive or intractable and thus are not applicable in real-world applications [8]. The recently emerged evidential deep learning (EDL) is a cost-effective approach, designed to quantify uncertainty via a deterministic model [23, 35, 4, 16]. The core mechanism behind EDL is to transform the classification evidence into conjugate distributions over the traditional model predictions as its outputs. The ability of EDL is remarkable for its effectiveness in capturing aleatoric, epistemic, as well as distributional uncertainty by a single model and a single forward pass [39].

Despite the great promise of EDL on well-curated data, through a comprehensive performance study, we have found that EDL is highly likely to yield imprecise distributions when facing extreme class imbalance. This can be explained by the fact that EDL is optimised via the empirical loss, i.e., the averaged loss of all training samples, which leads to biased classification evidence with monitory classes under-represented. Consequently, EDL generates both inaccurate categorical predictions and unfair uncertainty estimations.

Motivated by the above observations, in this paper, we propose a hybrid approach combining data-level and algorithm-level strategies to alleviate the performance degradation when applying EDL on imbalanced data. Specifically, toward a task-agnostic framework, our solution is, firstly, to train the model with balanced data via randomly over-sampling the minority classes, and then calibrate the output distribution via the validation set in a post-hoc fashion. To validate the effectiveness

2022 Trustworthy and Socially Responsible Machine Learning (TSRML 2022) co-located with NeurIPS 2022.

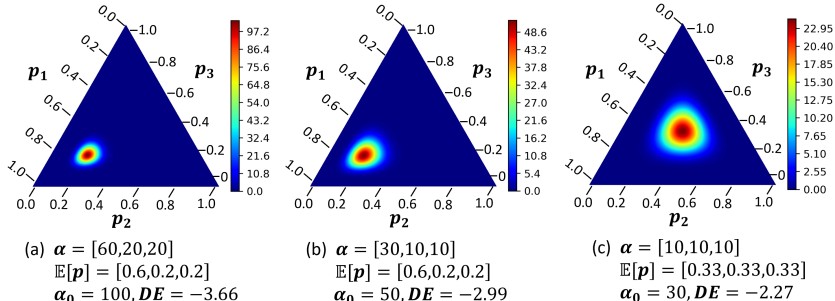

Figure 1: Three-class Dirichlet distribution. (a) and (b) point to the same predicted class but (b) has higher uncertainty indicated by the larger DE value. (c) shows an example with equal categorical probabilities across the three classes with the maximum DE value.

of our proposed Hybrid-EDL, we conduct extensive experiments on both artificial imbalanced and real-world healthcare datasets with label distribution skew. Results demonstrate the superiority of Hybrid-EDL compared to vanilla EDL and other baselines: Hybrid-EDL achieves less biased categorical prediction, better predictive confidence, and more precise out-of-distribution detection.

**Broader Impact.** Uncertainty quantification is largely under-explored for real-world applications. In fact, most studies merely leverage benchmark data like CIFAR10 and MNIST [25, 16, 28, 31], leaving the effectiveness a question for real applications with small data and skewed label distribution. Our study aims to bring attention to the more difficult yet realistic settings, making uncertainty quantification more helpful in achieving trustworthy deep learning.

## 2  Evidential Deep Learning for Uncertainty Quantification

Existing widely adopted deep learning models use a final softmax layer to output categorical probabilities $\boldsymbol{p^{(i)}} = [p_1^{(i)}, p_2^{(i)}, ..., p_K^{(i)}]$, with $\sum_c^K p_c^{(i)} = 1$, for a given input $X^{(i)}$. The output $\boldsymbol{p^{(i)}}$ is a point estimation which cannot capture the *epistemic* uncertainty because the model is deterministic [8]. Besides, the probability yielded by softmax is usually over-confident as it always predicts a close set for any given inputs, while the real-world is open with unseen classes [37].

Different from the traditional DL, underpinned by the Bayesian rule, EDL generates a Dirichlet distribution – the natural conjugate posterior of categorical probability $\boldsymbol{p}^{(i)}$ (i.e., $\boldsymbol{p}^{(i)}$ can be regarded as a multinomial distribution) as its output [11, 26]. The posterior $\boldsymbol{q}^{(i)} = Dir(\boldsymbol{\alpha}^{(i)})$ is parameterised by $\boldsymbol{\alpha}^{(i)} = [\alpha_1^{(i)}, \alpha_2^{(i)}, ..., \alpha_K^{(i)}]$ for $K$ classes. Using a uniform prior $Dir(\mathbf{1})$, $\alpha_c^{(i)}$ can be derived by $\boldsymbol{\alpha}^{(i)} = \mathbf{1} + \boldsymbol{l}^{(i)}$, where $l_c^{(i)}$ represents the likelihood/evidence of the $i$-th sample for class $c$ [26]. The fundamental target of EDL is to estimate the classification evidence for a given sample.

The presence of the Dirichlet distribution in EDL enables a better-calibrated way of quantifying *epistemic* uncertainty compared to the traditional point estimation [23, 35]. Additionally, the expectation of probability $\hat{\boldsymbol{p}}^{(i)}$ presents the average predictive confidence which reflects the *aleatoric* uncertainty. EDL is able to capture the *distributional shift* too: if no remarkable evidence can be modelled for a given input, the posterior $\alpha_c, \forall c \in K$ will approach 1. Overall, given an input $X^{(i)}$, an EDL model $f_\theta$ outputs distribution $\boldsymbol{q}^{(i)} = Dir(\boldsymbol{\alpha}^{(i)})$ with the predictive probability $\hat{\boldsymbol{p}}^{(i)}$, categorical prediction $\hat{y}^{(i)}$ and uncertainty measurement *Differential Entropy* ($DE$) inferred as below,

$$
\begin{aligned}
\hat{p}_c^{(i)} &= \mathbb{E}[p_c^{(i)}] = \frac{\alpha_c^{(i)}}{\alpha_0}, \\
\hat{y}^{(i)} &= \arg\max_c \mathbb{E}[p_c^{(i)}], \\
DE &= \mathbb{E}_{\boldsymbol{p} \sim \boldsymbol{q}}[-In(\boldsymbol{p})],
\end{aligned}
\tag{1}
$$

where $\alpha_0 = \sum_c^K \alpha_c^{(i)}$. $DE$ reflects how the energy is distributed, i.e., the "peakedness", in the Dirichlet distribution. A larger value corresponds to a higher uncertainty (see Figure 1).

Overall, EDL accomplishes uncertainty-aware classification by leveraging neural networks to estimate the distribution $\boldsymbol{q}$ instead of the probability $\boldsymbol{p}$. In this paper, we focus on *Posterior Network* [4],

a implementation of EDL, which leverages class-conditional normalising flows to generate the parameters of the Dirichlet distribution based on the feature extracted by common neural network. It is optimised by the loss function formulated as below [4],

$$\min_{\theta} \quad \mathcal{L} = \frac{1}{N} \sum_{i}^{N} \mathbb{E}_{\boldsymbol{p}^{(i)} \sim \boldsymbol{q}^{(i)}}[CE(\boldsymbol{p}^{(i)}, y^{(i)})] - \lambda \cdot H(\boldsymbol{q}^{(i)}), \tag{2}$$

where $CE$ denotes the cross-entropy loss with $CE(\boldsymbol{p}^{(i)}, y^{(i)}) = -\sum_{c} y_c^{(i)} \log p_c^{(i)}$ and $H$ denotes the entropy of $\boldsymbol{q}^{(i)}$. This loss approximates the posterior Dirichlet distribution for the true categorical distribution. Specifically, the first term, the expectation of the cross entropy, maximises the classification accuracy for the observed data, and the second term regularises the entropy, smoothing the distribution $\boldsymbol{q}^{(i)} = Dir(\boldsymbol{\alpha}^{(i)})$ to avoid overconfidence. $\lambda$ represents the weight used to trade-off between the two terms.

## 3 Performance Study of EDL

In this section, we analyse how class imbalance affects EDL's performance. Without the loss of generalisation, we use a binary classification task as an example.

### 3.1 Pitfalls of Loss Function

The function in Eq. (2) is an empirical loss designed to perform well on the average loss of $N$ training samples [14, 21]. Optimising this loss aims to minimise the negative log-likelihood of the ground-truth class (i.e., maximise the corresponding likelihood). For ease of analysis, we re-write Eq. (2) for the binary case as,

$$\mathcal{L} = \frac{1}{N} [- \sum_{y^{(i)}=1} P_1(X^{(i)}) - \sum_{y^{(i)}=2} P_2(X^{(i)}) + \sum_{i}^{N} F(X^{(i)})], \tag{3}$$

where $P_1$ and $P_2$ are the expected class probabilities for the two classes, respectively, with $P_1(X^{(i)}) = \mathbb{E}_{\boldsymbol{p}^{(i)} \sim \boldsymbol{q}^{(i)}}[\log p_1^{(i)}]$, $P_2(X^{(i)}) = \mathbb{E}_{\boldsymbol{p}^{(i)} \sim \boldsymbol{q}^{(i)}}[\log p_2^{(i)}]$, and $F(X^{(i)}) = -\lambda \cdot H(\boldsymbol{q}^{(i)})$[1].

By minimising Eq. (3), we hope to approach the estimated class likelihood to the true distribution. *We now show that with the extreme class imbalance, minimising the empirical loss may cause the minority class under-represented.*

Supposing we sample $N_1$ and $N_2$ ($N_1 + N_2 = N$ and $N_1 \gg N_2$) data points from the two class regions $\mathcal{R}_1$ and $\mathcal{R}_2$ for training, respectively. With a reasonable amount of training samples, the loss (Eq. (3)) can be derived as below,

$$\mathcal{L} = -\frac{1}{N} [N_1 \cdot \frac{1}{N_1} \sum_{X^{(i)} \in \mathcal{R}_1}^{N_1} P_1(X^{(i)}) + N_2 \cdot \frac{1}{N_2} \sum_{X^{(i)} \in \mathcal{R}_2}^{N_2} P_2(X^{(i)})],$$

$$= -\frac{1}{N} [N_1 \cdot \mathbb{E}_{\mathcal{R}_1}[P_1(X^{(i)})] + N_2 \cdot \mathbb{E}_{\mathcal{R}_2}[P_2(X^{(i)})]], \tag{4}$$

with $\mathbb{E}_{\mathcal{R}_c}$ denoting the expectation on class c. Because $N_1 \gg N_2$, $\mathbb{E}_{\mathcal{R}_1}[P_1(X^{(i)})]$ has a higher weight (i.e., $N_1/N$) in the loss. As illustrated in Figure 2, $P_1(X^{(i)})$ is more likely to be optimised faster than $P_2(X^{(i)})$ and it might present a larger value than the one (denoted by $P_1^*(X^{(i)}$ - dash line in Figure 2) optimised on balanced data (i.e., a larger $P_1$ corresponds a smaller $L$). As a result, the classification boundary will be shifted towards class 2. *Taken together, with the monitory class under-represented, both the prediction and uncertainty quantification of EDL might be unfair.*

### 3.2 Synthetic Data Analysis

To further validate the above analysis, we show some results on an synthetic binary classification case. We generate two linearly separable clusters with a sample ratio of 4:1. We feed those training samplings (80% of the data) into an EDL, and use the rest (20%) for validation. We also generate a testing set with the data densely covering the whole region to visualise the classification boundary

---

[1]We omit this regularisation term in the following analysis, as it is small and will not cause a difference between class 1 and 2.

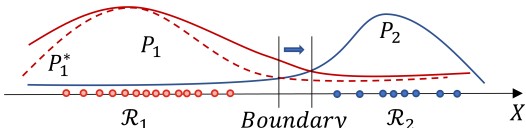

Figure 2: Estimated likelihood for classification with imbalanced data. A larger likelihood $P_1$ on class 1 (region $\mathcal{R}_1$) is likely to cause the classification boundary to shift towards class 2 (region $\mathcal{R}_2$).

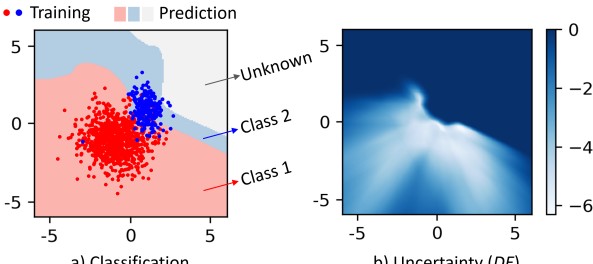

Figure 3: Visualisation for the binary classification study. a) Synthetic training samples (in dots) and categorical prediction results (in shades). b) Associated uncertainty measurement with a darker colour indicating that the prediction is more uncertain.

and the estimated uncertainty. As show in Figure 3(a), if $\alpha_1 > \alpha_2$ ($p_1 > p_2$), the point is predicted as class 1 and vice versa. When $\alpha_1 = \alpha_2$ ($\alpha$ is very close to 1), we label the class as unknown.

The pink shade (predicted category) covers most red data points of class 1. Correspondingly, as shown in Figure 3(b), the estimated uncertainty *DE* for class 1 is radially distributed, yielding an increasing uncertainty when the inputs gradually deviate from the centre of the training data. This behavior suggests that EDL achieves favourable performance for class 1 - the majority class. Yet, this is not the same for class 2, which is the minority. Firstly, the blue shade shows a relatively poor coverage of blue points in Figure 3(a): both class 1 and unknown class invade the region of class 2. Even for the correctly predicted area, the associated uncertainty is very high and even higher than the marginal area of class 1, as shown in Figure 3(b). This implies that the predictions are not confident and the estimated uncertainty is very high reaching the uncertainty for OOD samples. This again arises the concern that EDL could estimate unfair uncertainty for the minority class.

## 4   Our Solution: Hybrid-EDL

Built up the above findings, we propose Hybrid-EDL, a task-agnostic framework to improve EDL's performance when facing class imbalance. The core mechanism of our Hybrid-EDL is to first train the model with a conventional data balancing method and then further calibrate the model via a validation set to optimise the class-wise performance. The main new components of Hybrid-EDL are as bellow (with more details introduced in Appendix A),

**Training with Random Over-sampling.** In Hybrid-EDL, the class frequencies in the training set are first balanced by randomly reusing some samples from minorities [24]. Then deep learning model is fitted via the commonly used batch-based back-propagation [19].

**Post-hoc Calibration.** Although it is easy to achieve uniform class frequencies, enabling the empirical loss $\mathcal{L}$ to present a good approximation of the optimal loss, it additionally requires for the training set $\mathcal{D}_c$ to be representative of the true class region $\mathcal{R}_c$ for any class. Yet, generating high-quality data samples is non-trivial, and effective data-generating strategies are usually task-specific [36, 18]. To make the method generalised and effective, we further propose a novel post-hoc calibration module to compensate for the simple over-sampling. To target is to calibrate the classification evidence for minority classes by inspecting the class-wise performance via the validation set.

Specifically, for $K$ classes, we aim to find the weights $\boldsymbol{\omega} = [\omega_1, ..., \omega_c, ..., \omega_K]$ that can calibrate the parameters of the yielded Dirichlet distribution. We seek the best weights through a greedy search whose objective is to maximise the *unweighted average recall (UAR)* [34] on the validation set. For ease of notation, we assume class 1 is the majority while others are the minorities. Let $N_c$ denote the number of samples of class $c$ in the imbalanced training set. We initialise the search space

Table 1: Performance on CIFAR10 with various imbalance levels. The arrows after the metrics indicate the optimal direction. The best results are highlighted.

| | | ACC↑ | UAR↑ | REC$_{mi}$↑ | ECE↓ | AUC$_o$↑ |
|---|---|---|---|---|---|---|
| Balanced | EDL | 0.872 | 0.871 | - | 0.100 | 0.801 |
| Lightly Imbalanced | EDL | 0.830 | 0.830 | 0.822 | 0.134 | 0.780 |
| | Hybrid | 0.842 | 0.841 | 0.831 | 0.122 | 0.788 |
| Mildly Imbalanced | EDL | 0.763 | 0.764 | 0.749 | 0.166 | 0.650 |
| | Hybrid | 0.790 | 0.789 | 0.781 | 0.153 | 0.686 |
| Heavily Imbalanced | EDL | 0.698 | 0.700 | 0.668 | 0.213 | 0.627 |
| | Hybrid | 0.722 | 0.721 | 0.697 | 0.205 | 0.661 |

with $\omega_1 = 1$ and $\omega_c \in [1, \frac{N_1}{N_c}], (c > 1, N_1 > N_c)$. We then iteratively update UAR: for each input $X^{(i)}$ in the validation set $\mathcal{D}_v$, the prediction is made from $\boldsymbol{\alpha}^{(i)} \cdot \boldsymbol{\omega} = [\alpha_1^{(i)}\omega_1, ..., \alpha_c^{(i)}\omega_c, ..., \alpha_K^{(i)}\omega_K]$, according to Eq. (1). Therefore, for each $\omega_t$ in the search space, we can obtain a new *UAR*. Finally, the $\omega$ that leads to the highest *UAR* on the validation set will be chosen and used for inference.

## 5   Experiments and Results

We evaluate Hybrid-EDL from the three aspects: 1) **classification performance** by the overall accuracy (*ACC*), the mean of class-wise recall (*UAR*), and the averaged recall for minority classes (*REC$_{mi}$*), 2) **confidence estimation** by the expected calibration error (*ECE*) [10], and 3) **out-of-distribution detection** by the Area Under the receiver operating characteristic Curve (*AUC$_o$*) via using normalised *DE* to detect OOD data. We first use the benchmark data including CIFAR10 and SVHN at a variety of imbalance levels. Besides, three real-world healthcare datasets with natural label skew are also utilised for experiments. Details can be found in Appendix B and C.

### 5.1   Results on Artificially Imbalanced Data

We first utilise the CIFAR10 with class imbalance of various degrees: a step distribution among classes is kept and we term the imbalance ratio as the ratio between the size of the largest class and the smallest class [13]. We create light (ratio=10), mild (ratio=50), and heavy (ratio=100) imbalance for training. In addition, SVHN is employed as the OOD data to obtain *AUC$_o$*.

Results are summarised in Table 1. Our Hybrid-EDL consistently outperforms the vanilla EDL in all metrics regardless of the imbalance level. When the training data becomes increasingly imbalanced, both the classification performance (see *ACC*, *UAR* and *REC$_{mi}$*) and uncertainty quality (refer to *ECE* and *AUC$_o$*) decline compared to the results yielded by the balanced training data. Nevertheless, the drops of our Hybrid-EDL are less significant. Specifically, although the vanilla EDL and Hybrid-EDL achieve competitive performance when the imbalance is light, predictive confidence of our Hybrid-EDL is more reliable because we reduce *ECE* from 1.34 to 1.22 with an improvement of 9%. For the mild and heavy imbalance scenarios, Hybrid-EDL improves EDL by 3%∼5% for all metrics: showing higher classification accuracy, fairer predictive confidence, and better awareness of distributional shift. The extreme class imbalance is a very challenging task [1], while our hybrid method can effectively improve EDL with nearly no additional training cost and a very light post-hoc calibration module.

We also find that, as we hypothesised, the weight $\omega_c$ presents a larger value for smaller class. For example, in the mild balance case, we get $\omega_c$ up to 10 for the smallest class while $\omega_c = 1$ for the largest and second-largest classes. This suggests that the post-hoc calibration enhances the importance of the minorities, leading to less biased classifications.

### 5.2   Results for Real Applications

We conduct extensive experiments on another three real tasks with different data modalities and model backbones. For each task, we include two OOD datasets, named near OOD (i.e., with identical classes) and far OOD (i.e., with new classes), respectively. The data is as summarised in Table 2 Hybrid-EDL is compared to a series of traditional and start-of-the-art baselines (refer to Appendix D).

Table 2: A summary of real application datasets. #Train is the original training data size, which is split into training and validation folds with different seeds. #Test is the testing size. C is the number of classes and D is the input data dimension.

| Task | | | Dataset | | | | | | OOD Dataset | | | |
|------|---------|----------|-----------|--------|-------|---|------------------------------|-------------|------------|-------|----------|--------|
| Name | Backbone | Modality | Name | #Train | #Test | C | Ratio (%) | D | Near OOD | Size | Far OOD | Size |
| Task 1 | ResNet34 | Audio | ICBHI2017 | 4,274 | 2,641 | 4 | 52.8/27.0/12.9/7.3 | $1\times32,000$ | Stethoscope | 336 | ARCA23K | 2,264 |
| Task 2 | DenseNet121 | Image | HAM10000 | 7,206 | 2,809 | 7 | 67.1/11.1/11.0/5.1/3.3/1.4/1.1 | $3\times600\times450$ | ISIC2017 | 1,824 | CIFAR-10 | 10,000 |
| Task 3 | FCNet | ECG | EGC5000 | 4,500 | 500 | 5 | 58.4/35.3/3.9/2.0/0.5 | $1\times140$ | ECG200 | 200 | FetalECG | 1,965 |

Table 3: Performance comparison for real applications. The average results of five runs are shown. The best results are highlighted.

| | Task 1: Audio classification | | | | | | Task2: Image classification | | | | | | Task 3: Physiological signal classification | | | | | |
|---|------|------|------|------|------|------|------|------|------|------|------|------|------|------|------|------|------|------|
| | ACC↑ | UAR↑ | $REC_{mi}$↑ | ECE↓ | $AUC_o^n$↑ | $AUC_o^f$↑ | ACC↑ | UAR↑ | $REC_{mi}$↑ | ECE↓ | $AUC_o^n$↑ | $AUC_o^f$↑ | ACC↑ | UAR↑ | $REC_{mi}$↑ | ECE↓ | $AUC_o^n$↑ | $AUC_o^f$↑ |
| Softmax-FL | 0.578 | 0.399 | 0.286 | 0.366 | 0.573 | 0.580 | 0.845 | 0.712 | 0.699 | 0.122 | 0.699 | 0.913 | 0.922 | 0.710 | 0.646 | 0.073 | 0.681 | 0.746 |
| Softmax-AG | 0.599 | 0.411 | 0.291 | 0.342 | 0.583 | 0.595 | 0.869 | 0.733 | 0.705 | 0.107 | 0.734 | 0.945 | 0.934 | 0.725 | 0.663 | 0.068 | 0.706 | 0.771 |
| Mahalanobis | - | - | - | - | 0.644 | 0.672 | - | - | - | - | 0.736 | 0.990 | - | - | - | - | 0.788 | 0.951 |
| MCDP | 0.599 | 0.412 | 0.295 | 0.334 | 0.590 | 0.602 | 0.870 | 0.734 | 0.705 | 0.103 | 0.735 | 0.949 | 0.933 | 0.721 | 0.660 | 0.067 | 0.707 | 0.772 |
| Ensemble | 0.608 | 0.431 | 0.307 | 0.322 | 0.599 | 0.672 | 0.873 | 0.739 | 0.708 | 0.102 | 0.735 | 0.950 | 0.938 | 0.728 | 0.667 | 0.062 | 0.708 | 0.798 |
| EarlyExit | 0.607 | 0.430 | 0.300 | 0.321 | 0.598 | 0.670 | 0.871 | 0.733 | 0.707 | 0.104 | 0.736 | 0.983 | 0.936 | 0.726 | 0.665 | 0.062 | 0.710 | 0.810 |
| Bagging | 0.605 | 0.412 | 0.297 | 0.322 | 0.585 | 0.661 | 0.712 | 0.705 | 0.655 | 0.157 | 0.634 | 0.787 | 0.910 | 0.557 | 0.452 | 0.337 | 0.679 | 0.732 |
| Hybrid-EDL | 0.610 | 0.442 | 0.340 | 0.284 | 0.733 | 0.832 | 0.855 | 0.750 | 0.722 | 0.102 | 0.744 | 0.990 | 0.940 | 0.743 | 0.672 | 0.059 | 0.816 | 0.958 |

Result comparisons are summarised in Table 3. Overall, for both classification performance in terms of *ACC*, *UAR*, and $REC_{mi}$, and the quality of uncertainty as measured by *ECE*, $AUC_o^n$ and $AUC_o^f$, our proposed Hybrid-EDL outperforms the compared baselines for all the three tasks, except *ACC* on Task 2. Specifically, Hybrid-EDL improves *UAR* by 1.5%~2.6% and $REC_{mi}$ by up to 10.7% against the best baseline for all tasks. Hybrid-EDL also significantly reduces *ECE* for Task 1 and 3, and achieves comparable *ECE* on Task 2 with ensemble approaches. This suggests Hybrid-EDL enables fairer confidence estimation. In terms of using uncertainty to detect distributional shifts, Hybrid-EDL demonstrates a superior performance for the near OOD detection, with $AUC_o^n$ improved by 13.8%, 1.1% and 3.5% for the three tasks, respectively. For the far OOD detection, Hybrid-EDL shows competitive performance against the best baselines for Task 2 and 3, while for Task 1 which is more challenging, we achieve an improvement of 23.8%. Ablation study additionally suggests that each element of Hybrid-EDL provides an independent performance gain across all metrics against the vanilla EDL (refer to Appendix E.0.2).

**Implications.** Leveraging uncertainty for reliable application brings practical value. Firstly, corresponding to the lowest *ECE* in Table 3, Hybrid-EDL calibrates probabilities: the probability can better reflect the likelihood of a true prediction. Therefore, the predicted probability can be used as the confidence to filter out some uncertain predictions. The evidence shows if predictions with confidence under a threshold are excluded, the *ACC* climbs steadily as the increase of threshold (refer to Figure 5 in Appendix E). In addition, the uncertainty measurement *DE* can help to identify undesired inputs, corresponding to the promising $AUC_o^n$ and $AUC_o^f$ in Table 3 (with the distribution of $DE$ shown in Figure 6 in Appendix E). The in-domain validation and testing set present similar uncertainty distribution, yet the OOD sets show different patterns with a higher median value of *DE*. In this regard, a threshold according to the quantile of *DE* in the validation set can be selected: with *DE* larger than this threshold, an input is highly suspicious of an OOD sample. Overall, Hybrid-EDL allows quantified uncertainty estimation for real-world applications with skewed label distribution. The fair estimation can lead to more principled decision-making and enable deep learning models to automatically or semi-automatically abstain on samples for which there is high uncertainty, which could also help engender trust with domain experts and model users.

# 6   Conclusion

In this paper, we have theoretically and empirically demonstrated vanilla EDL's limitations in classification and uncertainty estimation in the presence of class imbalance. We proposed a Hybrid-EDL framework which combines training-phase data augmentation and validation-phase calibration to eliminate the bias inherited from the skewed training data distribution. Our experiments show that data augmentation is effective and that the post-hoc calibration can further boost performance, particularly for minority classes. Our study paves the way for more practical and reliable deployment of uncertainty-aware deep learning in the real world. In the future, calibrating the model without depending on an extra validation set will be studied.

## Acknowledgements

This work was supported by ERC Project 833296 (EAR). We also thank Nokia Bell Labs for their donation to the Centre of Mobile, Wearable Systems and Augmented Intelligence, Cambridge.

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

# Appendix

## A  Hybrid-EDL

To address the class imbalance problem, we propose Hybrid-EDL, a task-agnostic framework, to improve EDL's performance in categorical prediction and uncertainty quantification. The core mechanism of our Hybrid-EDL is to first train the model with a conventional data balancing method and then further calibrate the model via a validation set to optimise the class-wise performance. The framework is shown in Figure 4, and the procedure is illustrated in Algorithm 1.

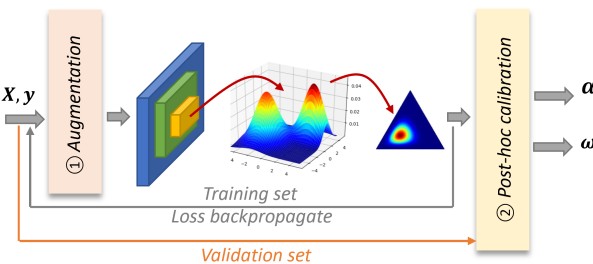

Figure 4: Hybrid-EDL overview. The model is trained with random over-sampling augmentation, and a post-hoc calibration is achieved via the validation set to find the best weights $\boldsymbol{\omega}$ to adjust the output $\boldsymbol{\alpha}$.

---

**Algorithm 1:** Hybrid-EDL Algorithm

---

**Data:** Training set $\mathcal{D}$, validation set $\mathcal{D}_v$
**Result:** Evidential model $f_\theta$, Calibration weights $\boldsymbol{\omega} = [\omega_1, .., \omega_K]$ for $K$ classes
1  //Training with random over-sampling.
2  Generate a balanced set $\mathcal{D}'$ from $\mathcal{D}$
3  **for** *epoch in [1,2,...,$Epoch_{max}$]* **do**
4  $\quad$ *Update $\theta$ by back-propagating $L(\mathcal{D}')$*
5  $\quad$ **if** *Accuracy of $\mathcal{D}_v$ stops increasing* **then**
6  $\quad\quad$ *Break*
7  $\quad$ **end**
8  **end**
9  *//Post-hoc calibration.*
10  *Initialise grid search space $W$ for $\boldsymbol{\omega}$.*
11  *Initialise the best $UAR_{best} = 0$, $\boldsymbol{\omega} = [1,1,...1]$.*
12  **for** *each $\boldsymbol{\omega}_t$ in $W$* **do**
13  $\quad$ *Initialise prediction list $\hat{Y}$ and label list $Y$*
14  $\quad$ **for** *each $(X^{(i)}, y^{(i)})$ in $D_v$* **do**
15  $\quad\quad$ $\boldsymbol{\alpha}^{(i)} = f_\theta(X^{(i)})$
16  $\quad\quad$ $\hat{y}^{(i)} = \arg\max_c \boldsymbol{\alpha}^{(i)} \cdot \boldsymbol{\omega}_t$
17  $\quad\quad$ *Add $\hat{y}^{(i)}$ to $\hat{Y}$ and add $y^{(i)}$ to $Y$*
18  $\quad$ **end**
19  $\quad$ *Calculate UAR by $\hat{Y}$ and $Y$*
20  $\quad$ **if** *UAR>$UAR_{best}$* **then**
21  $\quad\quad$ $\boldsymbol{\omega} \leftarrow \boldsymbol{\omega}_t$
22  $\quad$ **end**
23  **end**
24  *Return $\theta$ and $\boldsymbol{\omega}$*

---

# B  Evaluation Metrics

The evaluation will be from three aspects:

- **Classification**: We report overall accuracy (*ACC*), the average of class-wise accuracy (*UAR*), and the averaged recall for minority classes ($REC_{mi}$).
- **Confidence**: Uncertainty-aware model can calibrate the model's confidence, i.e., the confidence should reflect the likelihood that the categorical prediction is correct. For this purpose, we use the expected calibration error (*ECE*) [10].
- **OOD detection**: EDL is expected to detect distributional shifts. Hence, we include some OOD data for testing and report the Area Under the receiver operating characteristic Curve ($AUC_o$) by using normalised *DE* as the indicator [4].

A detailed formulation is list as below,

*ACC.* Acc is the proportion of the correctly predicted samples, formulated by:

$$ACC = \frac{1}{N_{test}} \sum_{i}^{N_{test}} \mathbb{1}(\hat{y}^{(i)} = y^{(i)}), \tag{5}$$

where $\mathbb{1}(\cdot)$ is the indicator function.

*UAR.* UAR is the averaged accuracy per class, which equals the mean of the dialog of the confusion matrix. We denote this as:

$$UAR = \frac{1}{K} \sum_{c}^{K} ACC(\hat{y}^{(i)}|y^{(i)} = c). \tag{6}$$

$REC_{mi}$. We term the class with the largest number of samples as the majority, while others are minorities. $REC_{mi}$ is the averaged recall of all minority classes as follows,

$$REC_i = \frac{1}{K-1} \sum_{c \in Minority} ACC(\hat{y}^{(i)}|y^{(i)} = c). \tag{7}$$

*ECE.* Expected calibration error measures the expected difference (in absolute value) between accuracies and the predicted confidences on samples belonging to different confidence intervals. We used $M = 10$ universal bins to calculate ECE as follows:

$$ECE = \sum_{m}^{M} \frac{|B_m|}{N_{test}} |ACC(B_m) - conf(B_m)|, \tag{8}$$

where bin $B_m$ covers the confidence interval $(\frac{m-1}{M}, \frac{m}{M}]$. $ACC(B_m)$ and $conf(B_m)$ are the ACC and the average predictive confidence for the samples having the predictive confidence within $B_m$.

$AUC_o$. The area under the receiver operating characteristic (AUROC, shorted as AUC) is used to measure the performance of OOD detection. We treat this as a binary classification task: OOD set is a positive class while in-domain (ID) data is the negative class. We conduct min-max normalise on the mixed ID and OOD testing set for the uncertainty measurement (for EDL methods, we use $DE$, and for other baselines, we use $Entropy$), resulting in the normalised values ranging $[0, 1]$ as the OOD probabilities to calculate AUC.

*DE.* Differential entropy measures the expectation of the entropy for the categorical distribution sampled from $Dir(\boldsymbol{\alpha})$ (after post-hoc calibration in our Hybrid-EDL). DE of a given sample is formulated by

$$DE = \mathbb{E}_{Dir(\boldsymbol{\alpha})}[-In(\boldsymbol{p})] \tag{9}$$

$$= In\boldsymbol{B}(\boldsymbol{\alpha}) + (\alpha_0 - K)\psi(\alpha_0) - \sum_{c=1}^{K}(\alpha_c - 1)\psi(\alpha_c), \tag{10}$$

where $\boldsymbol{B}$ is the beta function and $\psi$ is the diagamma function. In practice, we use python *scipy* library to calculate it[2]. We finally reported the averaged DE across the testing test,

$$DE = \frac{1}{N_{test}} \sum_{i}^{N_{test}} DE(Dir(\boldsymbol{\alpha}^{(i)})). \tag{11}$$

---

[2]https://docs.scipy.org/doc/scipy/reference/generated/scipy.stats.dirichlet.html

*Entropy.* For baseline models, $Entropy$ of the categorical distribution is used as the measurement of uncertainty, which is formulated by,

$$Entropy = \frac{1}{N_{test}} \sum_i^{N_{test}} \sum_c^K -p_c^{(i)} \cdot \log(p_c^{(i)}). \tag{12}$$

## C Datasets and Architectures

The binary classification data was governed in a standard way in the two-dimensional space[3]. We feed those training samplings (80% of the data) into an EDL model consisting of two fully connected layers and two normalising flows with a depth of 6, and use the rest (20%) for validation. The testing set consists of $500 \times 500$ data points in the [-10,10] region.

We also experimented with a benchmark and three real-world tasks with various data modalities. The details are introduced below,

**Benchmark.** We used VGG16 on the CIFAR10 image dataset for classification [4].

- **(ID)** We used the original balanced testing set of 10,000 images and uniformly sampled another 5,000 images from the original training set for validation. The rest part was used for training, i.e., around 4500 images per class.
- **(OOD)** We employed the testing set (26,000 images) of SVHN database as an opening set for CIHAR10, as they contain totally different categories.

*Setting.* To simulate the imbalanced data distribution, we downsampled the training set with the class ratio of: 1:0.9:0.8:0.7:0.6:0.5:0.4:0.3:0.2:0.1 (light imbalance), 1:0.89:0.78:0.67:0.56:0.45:0.34:0.23:0.11:0.02 (mild imbalance), and 1:0.5:0.4:0.3:0.25:0.2:0.15:0.1:0.05:0.01 (heavy imbalance). We run the experiments five times by down-sampling different classes to report the average performance. All images are cropped to $32 \times 32$ before feeding into the model.

**Task 1: Respiratory audio classification.** We explored the state-of-the-art ResNet34-based acoustic model to distinguish abnormal lung sounds from healthy sounds [7].

- **(ID)** ICBHI 2017 Respiratory Challenge[4] published a dataset collected from multiple microphones and stethoscopes [33]. The total 6,898 samples from 126 patients cover four classes: normal lung sounds (52.8%), crackle only (27.0%), wheeze only (12.9%), and both crackle and wheeze (7.3%).
- **(Near OOD)** A similar audio dataset named Stethoscope consists of 336 normal, crackle, and wheeze ausio samples [6]. we used it as ICBHI's co-variate shift counterpart, as although this dataset covers the same pathology, it is collected from a 3M Littemann electronic Stethoscope, differing from ICBHI.
- **(Far OOD)** ARCA23K is a dataset of labelled sound events originating from *Freesound*, and each clip belongs to one of 70 typically audio classes including music, human sounds, animal sounds, etc[5]. We used the validation set containing 2,264 clips.

*Setting.* For the ID data, we followed the official patient-independent training and testing splits of the Challenge. Samples from 47 patients were used for testing, while for the rest of the patients, we randomly divided them into five folds and hold out one fold per running to conduct five-fold cross-validation. For all ID and OOD datasets, audio recordings were re-sampled to 4KHz and divided into 8s clips. The clips were then transformed to Mel-spectrograms as the inputs of the model. For data augmentation, concatenation-based spectrogram generation was applied to increase the size for abnormal clips in the training folds [7].

**Task 2: Skin lesion image classification.** A DenseNet121 based image classification model was used to detect skin lesions [29].

---

[3]https://scikit-learn.org/stable/auto_examples/classification/plot_classifier_comparison.html
[4]https://bhichallenge.med.auth.gr/
[5]https://zenodo.org/record/5117901#.YkCsRk3MJPY

- **(ID)** HAM10000[6] contains 10,015 determatoscopic skin tumour images taken from multiple devices and demographics [38]. Image size is 600×450. The skin condition is labelled as one of the following classes: melanocytic nevi (67.1%), melanoma (11.1%), benign keratosis-like lesion (11.0%), basal cell carcinoma (5.1%), actinic keratoses (3.3%), vascular lesion (1.4%), or dermatofibroma (1.1%).

- **(Near OOD)** Another skin lesion dataset with 2,000 high-resolution varied-size images published by ISIC 2017[6] was used [5]. It was collected by another institute with a varied device from HAM10000, therefore we regard it as the near OOD.

- **(Far OOD)** The image classification benchmark CIFAR-10 with 10 non-skin classes was utilised as the far OOD.

*Setting*. For ID data, 30% was held out as the testing set, and five-fold cross-validation was implemented: four-fifths of the remaining 70% of the data for training and one-fifth for validation per running. Image augmentation was conducted by slightly modifying the brightness of the images in the minority classes for training. Images in ISIC2017 datasets were resized uniformly to 767×1,022 before feeding into the model.

**Task 3: Heart signal classification.** An electrocardiogram (ECG) is a simple test that can be used to check the heart's rhythm and electrical activity. A one-dimensional convolutional neural network FCNet has been developed to detect cardiovascular diseases from ECG [2].

- **(ID)** ECG5000 is a 20-hour long one-channel ECG dataset, which has been split and interpolated into equal-length (140) heart beats[7]. It consists of five classes: 58.4% are normal, 35.3% have heat failure typed R-on-T, 3.9% PVC, 2.0% SP, and 0.5% UB.

- **(Near OOD)** Another dataset consisting of 200 ECG recordings with a length of 178 was used as the near OOD[8], because the data acquisition method is different from ECG5000 [27].

- **(Far OOD)** A non-invasive fetal ECG dataset consists of 1,965 heatbeats with a length of 750[9]. As electrodes were placed on the mother's abdomen, the ECG is usually of lower amplitude than the maternal's, and thus we used it as the far ODD dataset.

*Setting*. We utilised a subset of 500 samples in the ID ECG5000 datasets for testing, and split the rest into five folds uniformly for cross validation. We also up-sampled the monitory classes with replacement to re-balance the training class distribution.

**Architecture.** Following the architecture of *Posterior Network* [4], in this paper, we used pre-trained backbone models as provided by Pytorch. On top of the backbone model, fully connected (FC) layers sized 64 with spectral normalisation (SN) and batch normalisation (BN) were included in DEL to map the high-dimensional feature vector into the representation with a length of 64. Then, $K$ (class number) radial normalising flows with a depth of $d$ are leveraged to estimate the density. We used $d = 8, 8, 12, 10$ for benchmark, Task 1, 2, and 3 respectively. The weight of the entropy regularisation in Eq. (2) $\lambda$ is set to 10e-5.

# D   Baselines

**Baselines.**   We compare Hybrid-EDL with deterministic models and several state-of-the-art uncertainty-aware approaches. A deterministic model utilises *softmax* on the logits yielded by the backbone network to generate the predictive probabilities, which is a commonly used baseline [12]. In the presence of imbalanced training data, we implement deterministic models with focal loss [22] and carefully-designed task-specific data augmentation [7, 29, 2], named **Softmax-FL** and **Softmax-AG**, respectively. Regarding uncertainty quantification, we implement Monte Carlo Dropout method (**MCDP**) [8], deep ensemble learning (**Ensemble**) [17], and multi-head early exists ensemble (**EarlyExit**) [32] as baselines. Those methods presented promising results from the literature but they are not designed for imbalanced data. For a fair comparison, we implement them via the same task-specific data augmentation as **Softmax-AG**. We also include the bagging-based ensemble approach

---

[6]https://challenge.isic-archive.com/data/
[7]https://timeseriesclassification.com/description.php?Dataset=ECG5000
[8]https://timeseriesclassification.com/description.php?Dataset=ECG200
[9]https://timeseriesclassification.com/description.php?Dataset=NonInvasiveFetalECGThorax1

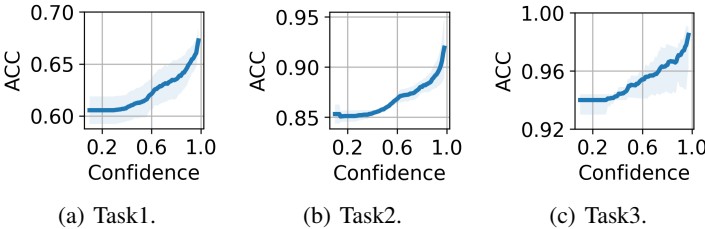

(a) Task1.        (b) Task2.        (c) Task3.

Figure 5: Performance of selective prediction: Accuracy (y-axis) on the testing sets with confidence above a certainty threshold (x-axis).

(***Bagging***) [40]. This method was proposed for imbalanced binary classification, but we extend it to the multi-class scenario. For the performance of OOD detection, built upon on ***Softmax-AG***, the state-of-the-art Mahalanobis distance-driven approach ***Mahalanobis*** that captures the feature distribution in the hidden space is employed as an additional baseline [20]. More implementation details can be found in Appendix C.

**Data Augmentation.** Baselines excluding *softmax-FL* and *Bagging* were implemented with task-specific data augmentation. In Task 1, audio samples were transformed into Mel-spectrograms with 64 bins. Data augmentation was implemented by concatenating two randomly selected Mel-spectrograms. For the crackle class, a Mel-spectrogram from normal and original crackle class were picked. Similarly, for the wheeze class, a Mel-spectrogram from normal and original wheeze class were used. For both crackle and wheeze classes, one crackle and one wheeze were mixed. After that, all spectrograms were divided into 8s clips. We used 8s because 80% of the samples have a length shorter than 8s. For those short ones, repeating padding was implemented. For Task 2, image augmentation was conducted by slightly modifying the brightness of the images in all classes excluding *melanocytic nevi* for training. We used OpenCV function with $\alpha = [0.9, 0.95], \beta = [0.05, 0.1]$[10]. For Task 3, as it is less challenging compared to other tasks proved by relatively high accuracy, over-sampling signals with added Gaussian noise were leveraged.

**Training.** Other implementation details as summarised below,

- *Softmax-AG*: Following the backbone model, two FC layers with Relu and Softmax activate function were used.
- *Softmax-FL*: When training the deterministic model, instead of using the general cross-entropy loss, the focal loss was applied with $\gamma = 2$. We adapted the implementation from this public repo[11].
- *MCDP*: A dropout rate of 0.5 is used for all models during training. This is changed to 0.2 for inference to avoid an accuracy drop. The inference is run five times per sample to get the averaged probabilities.
- *Ensemble*: We used five models to get the averaged probabilities.
- *EarlyExit*: We use the exits at the last five layers of the deterministic model.
- *Bagging*: $N_b$ is the size of the smallest class for training. For Task 3, it is smaller than 100. So we use $N_b = 1000$ with over-sampling of the minority classes, otherwise, the training size is too small.
- *Mahalanobis*: We used the feature from the last five layers to obtain Gaussian distribution. Unlike the original Mahalanobis score where a linear regression model is fit to identify OOD samples, we assume OOD samples are not achievable during training. Instead. we use the Mahalanobis distances from the validations set to normalise the corresponding score in the testing set, and take an average of this distance as the final score.

For all the methods in this paper, we use a learning rate of 10e-5, an optimiser of Adam, a batch size of 64, and a maximum epoch of 200. The best model is saved by the highest ACC on the validation set.

---

[10]https://opencv-laboratory.readthedocs.io/en/latest/nodes/core/convertScaleAbs.html
[11]https://github.com/gazelle93/Multiclass-Focal-loss-pytorch

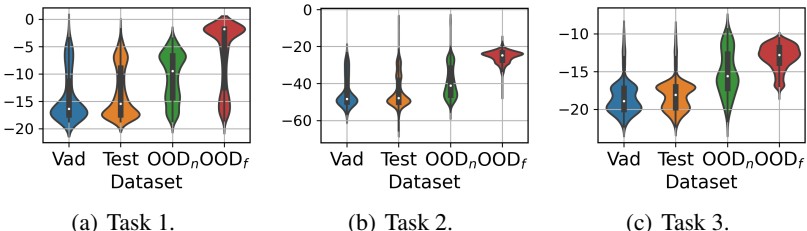

|  | (a) Task 1. | (b) Task 2. | (c) Task 3. |

Figure 6: Distribution of $DE$. Validation and testing sets have very similar and small $DE$s, while near and far ODD sets show larger $DE$s.

Table 4: Ablation study performance. **+ROS** denotes that we implement the random over-sampling during training, and **+PC** means the post-hoc calibration is conducted based on the vanilla EDL.

|  |  | ACC↑ | UAR↑ | $REC_{mi}$↑ | ECE↓ | $AUC_o^n$↑ | $AUC_o^f$↑ |
|---|---|---|---|---|---|---|---|
| **Task 1** | EDL | 0.591 | 0.268 | 0.119 | 0.304 | 0.655 | 0.734 |
|  | EDL+ROS | 0.616 | 0.434 | 0.316 | 0.297 | 0.700 | 0.768 |
|  | EDL+PC | 0.601 | 0.428 | 0.312 | 0.302 | 0.689 | 0.747 |
|  | Hybrid-EDL | 0.610 | 0.442 | 0.340 | 0.284 | 0.733 | 0.832 |
| **Task 2** | EDL | 0.688 | 0.601 | 0.575 | 0.214 | 0.688 | 0.803 |
|  | EDL+ROS | 0.860 | 0.735 | 0.712 | 0.105 | 0.701 | 0.896 |
|  | EDL+PC | 0.798 | 0.722 | 0.701 | 0.150 | 0.700 | 0.875 |
|  | Hybrid-EDL | 0.855 | 0.750 | 0.722 | 0.102 | 0.744 | 0.990 |
| **Task 3** | EDL | 0.914 | 0.317 | 0.237 | 0.240 | 0.786 | 0.887 |
|  | EDL+ROS | 0.940 | 0.690 | 0.622 | 0.062 | 0.790 | 0.920 |
|  | EDL+PC | 0.915 | 0.647 | 0.588 | 0.172 | 0.789 | 0.902 |
|  | Hybrid-EDL | 0.940 | 0.743 | 0.672 | 0.059 | 0.816 | 0.958 |

ResNet-34 and DenseNet-121 are initialised by the pre-trained checkpoints while other parameters are randomly initialised. All models are implemented by Pytorch 1.16 and we trained the models by a single Nvidia GPU with 64G memory. Codes will be publicly available once the paper is accepted.

# E  Additional Results for Real Applications

### E.0.1  Implications of Uncertainty.

We further explain how to leverage Hybrid-EDL for reliable predictions. First, corresponding to the lowest *ECE* in Table 3, Hybrid-EDL calibrates probabilities: the probability can better reflect the likelihood of a true prediction. Therefore, we are able to use the predicted probability as the confidence to filter out some uncertain predictions. As shown in Figure 5, if we reject the predictions with confidence under a threshold, the *ACC* climbs steadily as the increase of threshold. In addition, Hybrid-EDL can identify undesired inputs via the uncertainty measurement *DE*, corresponding to the promising $AUC_o^n$ and $AUC_o^f$ in Table 3. For this purpose, we showcase the distribution of $DE$ in Figure 6. It can be observed that the in-domain validation and testing set present similar uncertainty distribution, while the OOD sets show different patterns with a higher median value of $DE$. Therefore, we are able to choose a threshold according to the quantile of $DE$ in the validation set: with $DE$ larger than this threshold, an input is highly suspicious of an OOD sample.

Overall, in practice when deploying Hybrid-EDL, we can leverage $DE$ to filter out-of-distribution inputs, and then for the rest, we could utilise the probabilities to make confidence-aware predictions.

### E.0.2  Ablation Study.

Here, we study the individual components in our Hybrid-EDL framework and their influence on the final performance. Table 4 summarises the ablation study results. As can be seen, each element provides an independent performance gain across all metrics against the vanilla EDL. It is also noted that random over-sampling can effectively improve the overall accuracy and uncertainty estimation. Yet, as we hypothesised before, the model can still be biased because of the poor data coverage of the

minority classes. This can be validated by the notable gap between *ACC* and *UAR* of EDL+ROS: 12.5%∼25.0 for the three tasks. When only post-hoc calibration is adapted, although the improvement in terms of all metrics is not as good as those of EDL+ROS, the gap between *ACC* and *UAR* can be reduced to 7.6%∼23.8%. Consequently, applying post-hoc calibration on the model trained with over-sampling augmentation can compensate for the underestimated evidence for the minority classes, leading to a better trade-off between classification and uncertainty estimation.

