# OpenReview forum: "Hybrid-EDL: Improving Evidential Deep Learning for Uncertainty Quantification on Imbalanced Data"
_NeurIPS.cc/2022/Workshop/TSRML — TSRML2022_

### Official Review · Reviewer_7VFs · 2022-10-20
**Recommend acceptance**

**Overall Recommendation:** Recommend acceptance
**Overall Rating:** 8

**Summary:**

This paper studies uncertainty quantification on imbalanced data. By studying the behavior of Evidential Deep Learning (EDL), the authors propose Hybrid-EDL, which integrates data over-sampling and post-hoc calibration. Experiments demonstrate that Hybrid-EDL is more robust than the traditional approach.

**Strengths:**

(1) This paper is very well-written and easy to follow.
(2) This paper studies an important problem of imbalanced data on uncertainty quantification, which might be of general interest.
(3) The problem and the solution are well explained and I am convinced that the Hybrid-EDL is useful for solving data imbalance.
(4) Experiments well demonstrate the effectiveness of Hybrid-EDL.

**Weaknesses:**

N/A

**Review Confidence:**

3: The reviewer is fairly confident that the evaluation is correct

---

### Official Review · Reviewer_7mMp · 2022-10-20
**Review for paper Hybrid-EDL**

**Overall Rating:** 7

**Summary:**

This paper is motivated by the imprecise distribution generated by Evidential Deep Learning (EDL) when extremely imbalanced classes exist in the dataset. This paper proposes Hybrid-Evidential Deep Learning (HYbrid-EDL), which integrates over-sampling of the minority classes and post-hoc calibration using the validation dataset. Through extensive experiments, this paper shows that Hybrid-EDL improves accuracy in real-world healthcare datasets that contain skewed label distributions.

**Strengths:**

 - The paper is motivated by realistic problems: the imbalanced dataset in the real world widely exists, and the existing EDL method may yield imprecise distribution.
 - The experiments are well-designed and show that the proposed Hybrid-EDL has stronger performance than EDL along with a more imbalanced dataset.
 - The experiments are both conducted in artificially designed and real-world audio and image datasets.


**Weaknesses:**

 - The limitation is also included in the conclusion: The performance of HDL may heavily depend on the validation dataset.

**Overall Recommendation:**

The proposed Hybrid-EDL is practical when the validation dataset is available and of high quality. I would recommend accepting.

**Review Confidence:**

3: The reviewer is fairly confident that the evaluation is correct

---

### Official Review · Reviewer_V1Fo · 2022-10-21
**Review for paper30**

**Overall Rating:** 6

**Summary:**

The paper studies the special treatment of evidential deep learning when it is employed to quantify uncertainty on imbalanced data. The paper firstly motivates the necessity of EDL by analyzing the binary loss function when the number of samples in different classes are unbalanced. With the insights taken from the toy example, the authors put forward a solution called hybrid-EDL to add over-sampling of minority classes and post-hoc calibration into the traditional EDL.

**Strengths:**

1. The performance study shown in Section 3 appropriately shows the key challenge of EDL in unbalanced data and well motivates the proposed method.
2. The proposed method is simple but shown to be effective in various real-world tasks such as audio classification, image classification, and ECG signal classification.

**Weaknesses:**

1. It is not clear whether the compared baselines are strong enough and whether the comparison is fair enough. The reviewer cannot evaluate this point because the key difference between the proposed method and other baselines is not highlighted.
2. The proposed method seems to be standard in learning with long-tail data. It is not clear whether this idea is novel enough in the context of EDL.


**Overall Recommendation:**

The authors proposed a reasonable method and showed that the method can achieve competitive performance wrt existing baseline methods. The proposed method is not entirely novel since it seems to be standard in learning with long-tail data.

**Review Confidence:**

3: The reviewer is fairly confident that the evaluation is correct

---

### Decision · Program_Chairs · 2022-10-23

Accept